# Auditory cortex modulates call duration in rats
Wei Tang[1,3], Miguel Concha-Miranda[1,3] & Michael Brecht [1,2] ✉

The flexibility of human vocal production is well studied, but the direct role of the auditory system in regulating vocalizations in rodents remains largely unexplored. We show that during vocalizations, a fraction of rat auditory cortex neurons shows pre-call activity and different patterns of responses during calls and playback of calls. Additionally, we classified five auditory cortical vocalization responses: pre-call activated, onset activated, onset suppressed, ramping activated and ramping suppressed neurons. Intriguingly, onset suppressed cells can predict vocalization duration and occurrence. Injecting the auditory cortex with muscimol (GABA_A receptor agonist) prolonged vocalizations, while injecting the auditory cortex with gabazine (GABA_A receptor antagonist) shortened them. Similar reductions in call duration were observed during external white-noise stimulation of the auditory cortex and/or other auditory brain structures, resembling the effects of gabazine. Together, neuronal recordings, pharmacological interference and noise induced vocal modifications indicate direct modulation of vocal productions by the rat auditory cortex.

Vocalizations are intraspecific signals used by humans and many animal species. It plays a pivotal role in mediating the social interactions[1,2] and signaling emotions[3] in nearly all animal groups. Rodents produce ultrasonic vocalizations (USVs) as a means of expressing emotional states—both positive and negative—and communicating with conspecifics. Vocal production necessitates the synchronized functioning of phonation, articulation, and breathing, and engages a complex neuronal network that extends across the cerebral cortex, midbrain, and brainstem.

Findings from human studies have shown that the auditory cortex plays a key role in speech monitoring and control[4,5], while other studies on the vocalizations of deaf mice show that vocal behavior in rodents is not highly dependent on information processing in the auditory cortex or other auditory structures[6–8]. Interestingly, anatomical studies in rodents have shown that the auditory cortex has direct connections with brain regions involved in vocalization. For instance, it receives direct inputs from the motor cortical area[9,10], and layer V neurons in the auditory cortex send glutamatergic projections to lateral PAG neurons in mice[11].

These findings suggest that the auditory cortex may directly contribute to the regulation of vocal processing in rodents. However, we have only limited information as to whether the auditory cortex is directly involved in modulating self-generated vocalizations in rodents, or how motor and sensory information is integrated and processed within this region. Specifically, we aimed to address the following questions: Is the auditory cortex directly involved in encoding specific parameters of self-generated vocalizations in rats? If yes, does the auditory cortex play a causal role in regulating

specific vocal properties? Furthermore, can this effect be mimicked by activating the auditory cortex and/or other auditory brain regions through the application of external sound?

We present direct evidence for vocal modulation by rodent auditory cortex. The analysis of neuropixels recordings, pharmacological manipulations, sound recordings of calls, and different intensity levels of white noise offers strong evidence supporting the notion that neural ensembles in the auditory cortex may mediate vocal production in rats.

## Results

### Pre-call activity and differences between call and playback-evoked responses suggest auditory cortex activity is endogenously modulated during calls

In order to study the involvement of the auditory cortex in vocal production, we evoked sequences of vocalizations by electrically stimulating the rat periaqueductal gray (PAG) (AP − 6.5–7.0 mm; ML 0.75–0.8 mm; DV 4–4.5 mm). Consistent with our previous report[12,13], electrical stimulation (50–300 µA) consistently elicited calls within the frequency range from 20 to 35 kHz (Fig. 1A). We recorded the activity of auditory cortex cells (Fig. 1A, B), using high-density Neuropixels1.0 and 2.0[14] during call production. We additionally recorded 5-min period of the animal own calls, which were subsequently played back to the animal at the end of the experiment.

We identified 171 neurons in the auditory cortex from four rats (total recorded neurons per session: 25, 36, 54, and 56 neurons), across different cortical depth and aligned their activity (in 10 ms bins) to the onset and

[1]Bernstein Center for Computational Neuroscience Berlin, Humboldt-Universität zu Berlin, Berlin, Germany. [2]NeuroCure Cluster of Excellence, Humboldt-Universität zu Berlin, Berlin, Germany. [3]These authors contributed equally: Wei Tang, Miguel Concha-Miranda. ✉e-mail: michael.brecht@bccn-berlin.de

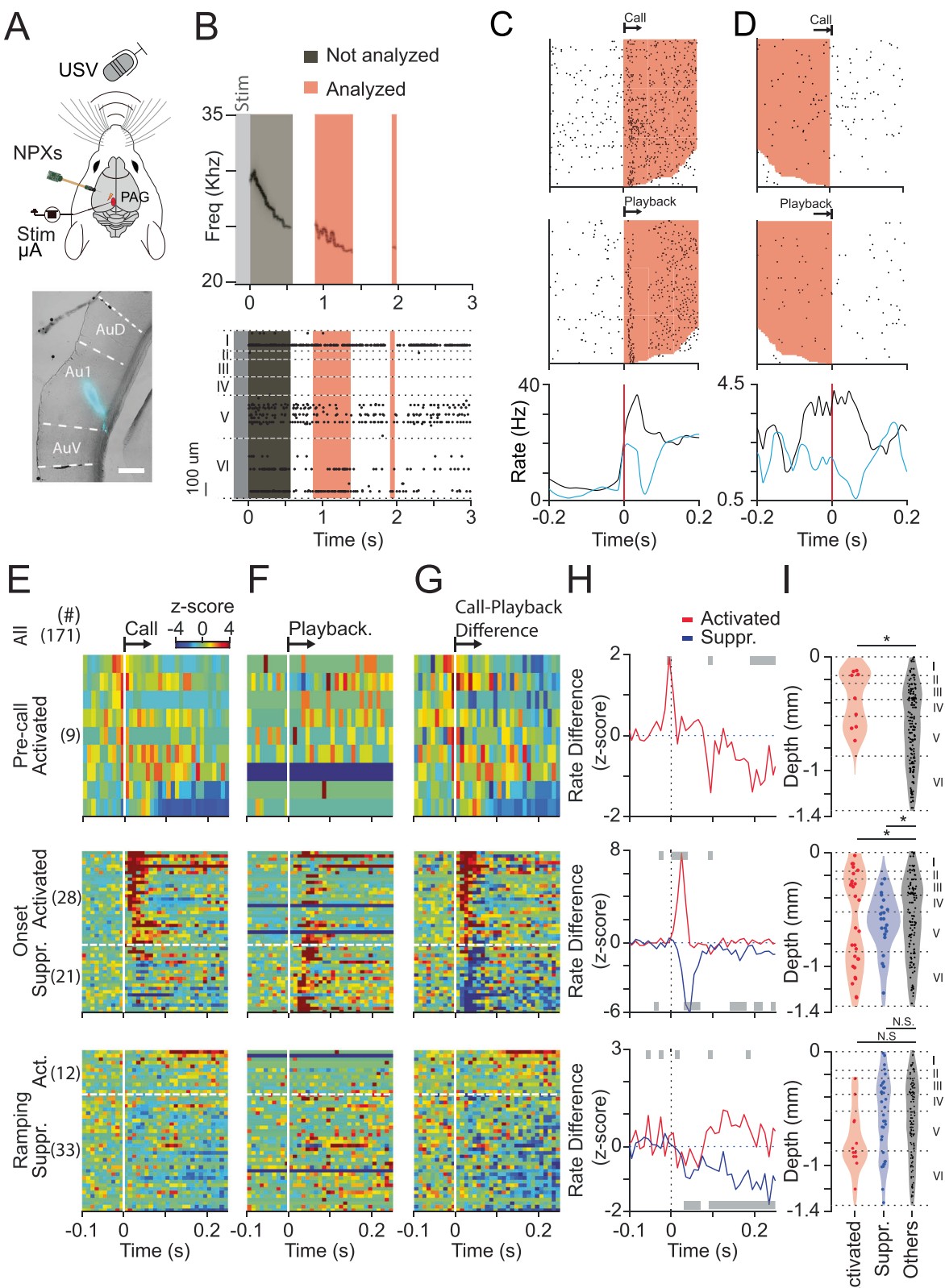

offset of calls and playback of calls (Fig. 1C). Neurons were then classified as pre-call activated (n = 9), onset activated (n = 28, Fig. 1C), onset suppressed (n = 21), ramping activated (n = 12, Fig. 1D), and ramping suppressed (n = 33) based on their activity patterns during call and/or playback (Fig. 1E–G). See Methods and Supplementary Fig. 1 for the classification procedure and criteria. Pre-call activated neurons were characterized by a

peak of activity before call onset (Fig. 1E top panel) that was not observed during playback (Fig. 1F, G top panels and Supplementary Fig. 2A). Onset activated (Fig. 1E middle panel, upper and Supplementary Fig. 2B) and suppressed neurons (Fig. 1E middle panel, lower and Supplementary Fig. 2C) showed a marked difference between call and playback onset (Fig. 1F, G, middle panels). Ramping cells displayed no clear onset response

**Fig. 1 | Depth-specific pre-call activity and differences between call and playback-evoked responses suggest auditory cortex activity is endogenously modulated.**
**A** Experimental design. Top: calls were elicited by electrically stimulating the PAG. Neural recordings were performed in the same hemisphere. Bottom: example histological image of probe track (Scale bar 500 um). **B** Top: electrical stimulation (200 ms duration) elicited a series of vocalizations. Dark gray: call not analyzed. Light red: calls analyzed. Bottom: auditory cortex neural activity was recorded using Neuropixels1.0 and 2.0. Raster plot of recorded neurons from one animal after PAG stimulation during the same calls showed in (**A**). Light gray area indicates electrical stimulation. Dark gray and red shaded areas indicate vocalizations. Calls occurring before 200 ms after stimulation offset were not included (dark gray area). In this example, the two later calls (red shaded areas) were included in the analysis. We estimated the location of all channels within the cortex. **C** Raster plot and PSTH of an example onset activated neuron aligned to the onset of calls or playback of calls (shaded red area indicates the presentence of calls or playback of calls). **D** Raster plot and PSTH of an example ramping activated neuron aligned to the offset of calls (shaded red area indicates the presentence of calls or playback of calls). **E** Population responses to call onset (vertical dashed lines) for different functional cell types. Each of the three panels shows the baseline normalized firing rates of neuron belonging to a particular cell-type (rows represent individual neurons). Neurons were classified as pre-call activated ($n = 9$), onset activated ($n = 28$; above dashed horizontal line), onset suppressed ($n = 21$; below dashed horizontal line), ramping activated ($n = 12$; above dashed horizontal line), and ramping suppressed ($n = 33$; below dashed horizontal line). See methods and Supplementary Fig. 1 for more details about the classification criteria and procedure. **F** Population responses of the same cells from **E** to the onset of playback. **G** Response dynamics resulting from subtracting playback responses (**F**) from call responses (**E**). Sorting of neurons in (**E–G**): pre-call neurons are sorted according to their response before call onset; onset-neurons are sorted according to their Call-Playback difference after call onset; ramping neurons are sorted according to their response before offset. Color code is shown on (**E**), top right. **H** Mean Call-Playback responses to call onset (vertical dashed line). Periods where call and playback responses are significantly different (paired t-test, $p < 0.05$) are indicated with gray bars. **I** Depth distribution of each cell type compared with the distribution of the rest of the auditory cortex neurons. The dashed line describes the putative cortical layers according to Smith[33]. As an example: the top panel shows the distribution of pre-call neurons ($n = 9$) compared with the distribution of the remaining neurons ("Others": $n = 171–9 = 162$) (* = $p < 0.05$).

but rather an increasing ramping activity (Fig. 1E–G lower panels and Supplementary Fig. 2D, E) that peaked during offset.

When comparing their responses between call and playback, pre-call activated neurons showed a significant difference during the 10 ms before call onset (Fig. 1H, upper panel); onset activated neurons showed a significant difference starting just after call onset that lasted 40 ms (Fig. 1H, middle panel, red), while onset suppressed neurons showed a significant difference starting 30 ms after call onset that lasted 40 ms (Fig. 1H, middle panel, blue). Ramping activated neurons showed multiple scattered periods where their call and playback responses were significantly different (Fig. 1H, lower panel, red), likely due to the low number of neurons used for comparison. Finally, ramping suppressed neurons showed a brief interval of significance at 30 ms and a persistent significant difference starting at 90 ms (Fig. 1H, lower panel, blue).

To further characterize the difference between call and playback, we performed an additional experiment where playback stimuli were presented at different intensities (Supplementary Fig. 2F–I). Changing the playback intensity between −30 dB and −10dB elicited a proportionate response on auditory cortex neurons. Playback at −30 dB elicited significantly weaker neural responses compared to playback at −10 dB (Supplementary Fig. 2F, G, Wilcoxon signed-rank test: $W = 364.5$, $z = −2.30$, $p = 0.02$) and −20 dB ($W = 302.5$, $z = −2.37$, $p = 0.006$). We didn't find a difference between the two loudest stimuli (signed rank 650.5, $z$: 0.64 $p = 0.52$), probably due to saturation of the neural response.

Neural responses to an animal's own calls can be influenced by both the sound of the calls and the brain's endogenous activity. This internal activity might cause faster responses, for example, due to anticipatory signals or to corollary discharge-like mechanisms. To test this possibility, we measure the response latency to both the animal own calls and playback stimuli, by calculating the time between call or playback onset and the first detected spike. This analysis was performed in a subset of our experiments (one animal; 56 recorded neurons, 31 of which were call-responsive), where we delivered playback stimuli at different decibel levels in a randomized block order. For this analysis, we only included neurons with low pre-call firing rates (≤10 Hz). The distribution of pre-call firing rates was clearly bimodal with a natural cut-off around 10 Hz. Restricting the analysis to these low-firing neurons helped minimize baseline activity as a potential confounding factor, as neurons with high firing rates before calls also exhibited lower firing rates prior to playback. Under these conditions, call responses had shorter latencies than all playback stimuli (Supplementary Fig. 2H), even though there were not rate differences in the 100 ms before call onset among the same groups (Supplementary Fig. 2I). This suggests that endogenous signals may shorten neural responses to calls.

We also analysed the laminar location of these neurons and found that functional types showed significant differences (Fig. 1G). Pre-call activated neurons were located more superficially than the remaining of neurons (mean depth: 0.38 mm, $p < 0.035$, $t = −2.13$, d.f. = 169). Onset activated neurons were distributed more eccentrically (mean eccentricity: 0.26 mm, $p < 0.027$, $t = 2.24$, d.f. = 148). Onset suppressed neurons were observed mainly in deep layers (mean eccentricity: 0.19 mm, $p < 0.024$, $t = −2.30$, d.f. = 141).

In conclusion, auditory cortex neurons show distinct characteristics in their pre-call activity, call-playback difference, and depth distribution, suggesting the auditory cortex is endogenously modulated during call production.

## Onset suppressed neurons predict call duration and call occurrence

We further studied the discharges of onset suppressed cells in vocal emission. The low firing rates of many of these cells (with 48% of cells with firing rates lower than 2 Hz) prevented the analysis of single neuron activity within the short time windows associated with calls (~100 ms). To address this, we took advantage of the fact that we recorded multiple onset suppressed cells in some of our experiments ($n = 2$, $n = 1$, $n = 12$, $n = 6$ for each experiment), and target our analysis to their population activity instead. At the population level, the activity of onset suppressed neurons showed a striking several-fold difference between playback and call (Fig. 2A). The population activity of onset suppressed neurons could predict the duration of the following calls based on their firing rate level within the 100 ms time window before call onset (Fig. 2B, C). The linear relationship remained highly significant even after removing any combination of one ($p < 2.10 \times 10^{-8}$) or two neurons ($p < 3.34 \times 10^{-5}$), indicating that the observed effect was not driven by a small subset of cells. The effect was independently observed in three different animals (Supplementary Fig. 3A, B), and showed the strongest effect among all other cell types (Fig. 2D, Supplementary Fig. 3C). Importantly, the effect was almost specific to onset suppress cells, since most other cells showed a non-significant correlation with only one cell type showing a significant correlation after multiple comparison correction (Supplementary Fig. 3C, Onset Activated cells, $r = 0.16$ and $p < 0.048$ Bonferroni corrected). We further tested if the activity of these cells could also predict the occurrence of calls. We took advantage of the variability in the number of calls evoked by PAG stimulation to investigate whether the activity of onset-suppressed cells after the offset of a certain call could anticipate the termination of a call sequence. We obtained two groups of calls of equivalent duration distribution (see methods and Supplementary Fig. 4), where no call was produced in the 250 ms interval after call offset. We divided calls that were followed by another call after this interval (Fig. 2E, red) and calls with no subsequent call after this interval (Fig. 2E, gray). Matching the duration distribution between these two call groups was important, given the strong correlation of this cell type with this call feature. We then defined the

**Fig. 2 | Onset suppressed neurons predict call duration and call occurrence. A** Population rate of all onset-suppressed neurons within one experiment, in response to the call (red) and playback (PB, black) onset. **B** Raster plot of the activity of all onset-suppressed neurons within one animal after the call onset. Calls are sorted according to their duration. Red shaded area indicates the presence of calls. **C** The Population Rate during the 100 ms window before call onset is highly correlated with the duration of the upcoming call (black line represents linear regression fit. *p*-value of the corresponding Fisher test is indicated on top). **D** Explained variance from the mixed linear model between the population rate of each cell type and call duration. Activity of each animal was pooled together by z-scoring the population rate and call duration within each animal. **E** Left: Call sequences can include different numbers of calls. Calls followed by another call are indicated in red. Calls not followed by another call are indicated in gray. Only calls with inter-call intervals longer than 250 ms and with matching call durations were selected. For each call, the predicting interval is defined as the 150 ms interval starting 100 ms after the call offset. See Supplementary Fig. 4 for a more detailed description of the call selection criteria. **F** Up: Using a Support Vector Machine model (SVM), the rate during the 100–250 ms predicting interval after each call was used to predict the occurrence (or not) of a call afterward. Bottom: Confusion matrix of the SVM model of one animal. **G** True positive and true negative rates of the trained SVM for the three analyzed animals.

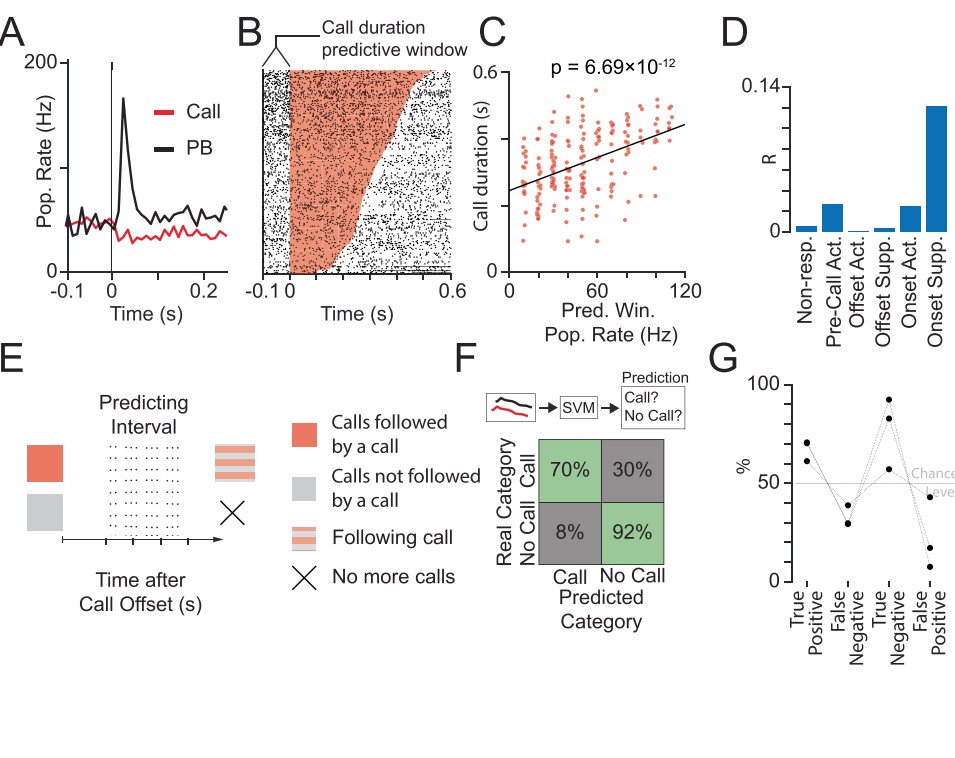

"predicting interval" as the 150 ms interval starting 100 ms after the call offset for both groups of calls (Fig. 2E, "predicting interval"). We then assessed the population activity of onset-suppressed neurons for calls with the following calls and for calls with no subsequent calls. The population rate of these cells during the predicting interval could predict the occurrence (or not) of another call. To achieve this, we trained a Support Vector Machine model using the binned rate during the predicting interval (20 ms bins). The trained model exhibited up to 92% accuracy in predicting the termination of a call sequence, and up to 70% accuracy in anticipating that another call is coming (Fig. 2F). All examined animals (*n* = 3) predicted above chance the occurrence of another call (Fig. 2G). In conclusion, onset-suppressed neurons can predict the duration and occurrence of calls.

## Auditory cortex is involved in modulating total call duration

In order to further investigate the direct involvement of the auditory cortex in vocal production, we suppressed neuronal activity during vocalization by locally administering the GABAA (γ-aminobutyric acid) receptor agonist muscimol into the auditory cortex (Fig. 3A and Supplementary Fig. 5A). The experiment consisted of three consecutive sessions, as elaborated in the methods section. The administration of saline had no discernible impact on either the total duration (Fig. 3B–D) or mean frequency of calls (and Supplementary Fig. 5B). In contrast, we found that the injection of muscimol into the auditory cortex led to a significant increase in total call duration (Fig. 3B–D), accompanied by a decrease in mean call frequency (Supplementary Fig. 5B) compared to the controls.

In subsequent experiments, we injected gabazine into the auditory cortex to activate auditory cortex neurons by blocking the GABAergic input through the GABA_A receptor (Fig. 3E and Supplementary Fig. 5C). Remarkably, the intra- auditory cortex application of gabazine (Fig. 3E–H) resulted in a significant reduction in total call duration compared to both the pre- and saline-sessions. However, the treatment of gabazine had no significant effect on the mean call frequency of animals (Supplementary

Fig. 5D). This observation might indicate that global activation of the auditory cortex might be necessary for regulating total call duration in rats within our behavioral paradigm.

## White noise reduces total call duration, along with changes in the number of calls, mean call frequency, and vocal intensity

To further understand the role of the auditory cortex in vocal production, we performed similar experiments in which we delivered white noise stimulation instead of pharmacologically stimulating neural activity. This approach has several advantages: noise stimulation may recruit (or inhibit) auditory cortex cells while preserving the network's excitatory-inhibitory balance. Additionally, white noise is a relatively more natural form of stimulation, whereby the auditory cortex is recruited through the auditory pathway. Discrepancies observed between the pharmacological stimulation and noise experiments may suggest the involvement of other auditory areas or indicate a role of the preserved excitatory-inhibitory balance in call production.

In order to assess the effect of white noise on vocal production, we evoked sequences of vocalizations while simultaneously delivering in-ear white noise of varying intensities. The experiment consisted of alternating sequences of vocalizations with white noise ("Noise" condition) and without white noise ("Baseline" condition). We started Noise trials by playing 7 s white noise stimuli; after 5 s from noise onset, electrical stimulation triggered a sequence of calls lasting less than 2 s, that is until before the noise stimulus ended (Fig. 4A). Each Noise condition was directly followed by a Baseline condition, which consisted of 5 s of baseline (no noise stimulus) followed by the electrical stimulation and a 2 s period for the call sequence to develop. Subsequently, each baseline condition was followed by another Noise condition. Under these conditions, applying the same stimulation protocol in a noisy environment led to shorter total call duration, higher call frequency, and increased vocal intensity (Fig. 4B–D; Supplementary Fig. 6).

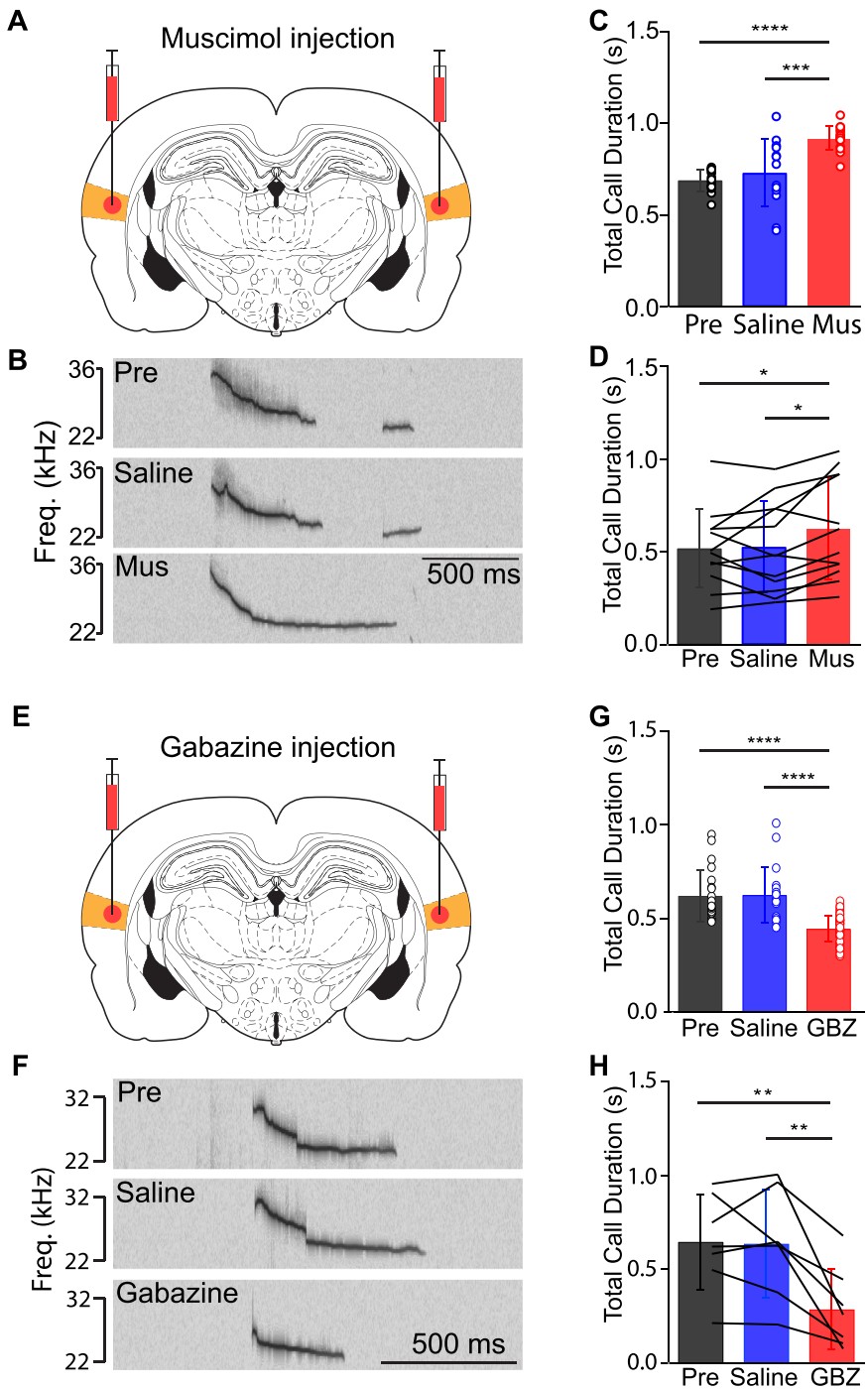

We then tested if the reduction in total call duration and the increase in call frequency were correlated with noise intensity. We repeated the same Noise-Baseline protocol while reducing the noise intensity in 10 dB steps. This resulted in a series of Noise and Baseline conditions of varying noise intensities, that were presented in a randomized order (0, −10, −20, −30, or −40 dB noise levels). We then compared the total call duration of each call sequence, using paired comparisons between consecutive noise and baseline trials (Fig. 4B). Louder noises produced stronger and more consistent differences in call duration, which were then associated with lower *p*-values. These effects were consistent across animals (Fig. 4C). We observed a significant, albeit less consistent, effect of increased call loudness during Noise conditions ("Lombard effect", Supplementary Fig. 6A), as well as a strong and consistent effect of noise on the reduction of the number of calls, and on the increase of call frequency (Supplementary Fig. 6B, C).

We then conducted rank correlations between the reductions in total call duration observed in each Baseline/Noise pair and the corresponding dB level of the trial (Fig. 4D). A substantial portion of the animals (80%, 8/10) exhibited a negative correlation between noise intensity and the reduction in total call duration. Additionally, we performed the same analysis on call loudness, number of calls, and call frequency. A modest majority of animals (60%, 6/10) showed a significant positive correlation between noise intensity and increases in call loudness (Supplementary Fig. 6D). However, only a minority of animals (40%, 4/10) showed a significant correlation between noise intensity and the reduction in the number of calls (Supplementary Fig. 6E), likely due to a floor effect on call number reduction. At higher noise intensity levels, once only one call was evoked, further increases in noise levels were primarily reflected in a reduction in call duration. Finally, most animals showed a positive correlation between noise intensity and call

**Fig. 3 | Local injection of muscimol and gabazine into rat auditory cortex bidirectionally modulate the total call duration.** **A** Schematic illustration of the coronal section of the rat brain, depicting the site of injection of the GABAA-receptor agonist muscimol in the auditory cortex. **B** Top: Spectrogram of calls from an example trial evoked by electrical stimulation in lPAG without any intervention in the auditory cortex (Pre). Middle: Spectrogram of calls from an example trial evoked by electrical stimulation in lPAG after the saline injection into the auditory cortex (Saline). Bottom: Spectrogram of call from an example trial evoked by electrical stimulation in lPAG after the muscimol injection into the auditory cortex (Mus). **C** The total call duration of each trial from the example animal in (**B**) that was measured before (Pre) and after the injection of saline (Saline) and muscimol (Mus) into the auditory cortex, respectively. A repeated measures one-way ANOVA showed a significant effect of the pharmacological manipulation [$F_{(2, 40)} = 17.05$, $P = 0.0000044$]; A post-hoc Tukey's test showed that the total call duration in the muscimol treatment was significantly longer than that in the pre-treatment ($p = 0.00000680$) and saline control treatments ($p = 0.00026392$) (*$p < 0.001$, **$p < 0.0001$). **D** The summary of the total call duration from all animals that were measured before (Pre) and after the injection of saline (Saline) and muscimol (Mus) into the auditory cortex, respectively ($n = 12$). A repeated measures one-way ANOVA showed a significant effect of the pharmacological manipulation [$F_{(2, 22)} = 5.697$, $P = 0.0101$]; a post-hoc Tukey's test showed that the total call duration in the muscimol group was significantly longer than that in the pre-treatment ($p = 0.0166$) and saline control treatments ($p = 0.0263$) (*$p < 0.05$). **E** A

schematic illustration of the coronal section of the rat brain, depicting the site of injection of the GABAA-receptor antagonist gabazine in the auditory cortex. **F** Top: Spectrogram of call from an example trial evoked by electrical stimulation in lPAG without any intervention in the auditory cortex (Pre). Middle: Spectrogram of call from an example trial evoked by electrical stimulation in lPAG after the saline injection into the auditory cortex (Saline). Bottom: Spectrogram of call from an example trial evoked by electrical stimulation in lPAG after the gabazine injection into the auditory cortex (GBZ). **G** The total call duration of each trial from the example animal in (**F**) that was measured before (Pre) and after the injection of saline (Saline) and Gabazine (GBZ) into the auditory cortex, respectively. A repeated measures one-way ANOVA showed a significant effect of the pharmacological manipulation [$F_{(2, 123)} = 45.18$, $P = 0.000000000000002$]; a post-hoc Tukey's test showed that the total call duration in the gabazine treatment was significantly shorter than that in the pre-treatment ($p = 0.00000004$) and saline control treatments ($p = 0.0000000002$) (****$p < 0.0001$). **H** The summary of the total call duration from all the animals that were measured before (Pre) and after the injection of saline (Saline) and gabazine (GBZ) into the auditory cortex, respectively ($n = 7$). A repeated measures one-way ANOVA showed a significant effect of the pharmacological manipulation [$F_{(2, 12)} = 10.81$, $P = 0.0021$]; a post-hoc Tukey's test showed that the total call duration in the gabazine group was significantly shorter than that in the pre-treatment ($p = 0.0040$) and saline control treatments ($p = 0.0049$) (**$p < 0.01$). Results are mean ± s.d.

frequency (Supplementary Fig. 6F). We also performed an equivalent analysis comparing each call, and we found similar effects.

In summary, 8 out of 10 animals showed an effect on total call duration and frequency proportional to noise intensity, but only 6 out of 10 animals increased call loudness proportional to noise intensity. In conclusion, compared with the baseline condition, noise stimuli strongly decreased the total call duration and number of calls, while markedly increasing call frequency and moderately increasing call loudness.

### During-call and pre-call effect of noise in call duration

To determine whether the effect of noise on call duration depends on stimulation timing relative to call onset, we designed an additional experiment with two conditions: (1) pre-call noise, where white noise ended just before the call onset, and (2) during-call noise, where noise began around the time of call initiation. We selected an animal that showed a strong and consistent effect of noise on vocalizations using the 7 s noise stimulus, including a clear effect on the first call alone (Supplementary Fig. 7A). Almost all noise trials (49/51) showed a decrease in call duration compared with baseline calls (Wilcoxon signed rank test, $z = 6.16$, $p < 7.35 \times 10^{-10}$) and all of them (51/51) showed an increase in call frequency (Wilcoxon signed rank test, $z = -6.12$, $p < = 5.14 \times 10^{10}$). We then presented white noise in both pre-call and during-call conditions (Supplementary Fig. 7B). In the pre-call condition, we obtained multiple trials where noise onset occurred 0–100 ms before call onset ($n = 34$). For each of these trials, we measured the difference in call duration and fundamental frequency between the first call in the baseline condition and the first call in the feedback condition. Most trials showed longer (Wilcoxon signed rank test, $z = 4.40$, $p < = 1.07 \times 10^{-5}$) and lower-frequency (Wilcoxon signed rank test, $z = -2.28$, $p < = 0.023$) calls during the baseline condition throughout the entire pre-call window (Supplementary Fig. 7D, E). In the feedback condition, we collected trials in which noise ended within ± 100 ms of call onset ($n = 42$). When the noise offset occurred within 100 ms before call onset, calls were longer (Wilcoxon signed rank test, $z = -2.385$, $p < 0.0086$) but showed no change in frequency (Wilcoxon signed rank test, $z = 0.44$, $p = 0.67$, Supplementary Fig. 7G, H). Conversely, when the noise offset occurred within 100 ms after call onset, call duration was unchanged (WilcoxPon signed rank test, $z = -0.71$, $p = 0.48$), but fundamental frequency increased (Wilcoxon signed rank test, $z = -2.32$, $p < 0.021$, Supplementary Fig. 7G, I).

In summary, white noise increases call length when delivered before call onset, but reduces it when delivered afterward—with the post-onset effect overriding the pre-onset effect when both are present (Supplementary Fig. 6A). Changes in frequency, however, occur only when noise is applied after call onset.

## Discussion

### Summary

In our present investigation, we evaluated the role of the auditory cortex in regulating vocal production in rats. By comparing the neuronal activity of the auditory cortex during call production and playback, we identified five different functional cell types in this region. Among these types of neurons, we discovered a specific cell type located in the deep layers of the auditory cortex that anticipates information about vocal duration and call occurrence. We provide both necessary and sufficient evidence demonstrating that bidirectional manipulation of the auditory cortex can directly and globally modulate the total duration of vocal emissions in rats within our behavioral paradigm. In the end, we found similar changes in total call duration by stimulating the auditory cortex and/or other auditory brain regions with external white noise, mimicking the effects observed in pharmacological stimulation experiments.

### Neuronal responses to vocalizations

Neurophysiological studies of the role of the auditory cortex in rodent vocalizations have traditionally been conducted from a sensory perspective. For instance, previous studies have shown that the primary auditory cortex in rats plays a role in processing rodent social communication and the sensory signals associated with USVs produced by conspecifics[15–17]. While neurophysiological studies on the cortical encoding of self-generated vocal-related features are relatively scarce in rodents. The most comparable studies were performed in marmoset monkeys, where Eliades and Wang conducted physiological experiments using single-unit recordings to study the neuronal activity of the auditory cortex during call emission and passive play-back of vocalizations[18,19]. Their main findings show that cells activated during vocal emission also exhibited activation in response to playback vocalizations, while cells inhibited during vocal emission showed a broader range of responses to playback stimuli, often tending to show excitation in response to playback. In agreement with their results, we demonstrate that some cells showing vocalization-related excitation also show increased firing when rats are exposed to playback of their own vocalizations (Fig. 1F). According to our knowledge, Eliades and Wang only reported the presence of two different types of neuronal responses, both inhibited and excited, in the auditory cortex during vocalization in marmoset[20]. Through comparing differences between call and playback-evoked responses, we were able to further identify five different cell types: Pre-call activated, onset activated, onset suppressed, ramping activated, and ramping suppressed. During self-produced vocalization, onset-activated cells show similar response dynamics to those observed in Eliades and Wang's study[19]. In addition, among our newly identified cell types, we found a specific type, termed

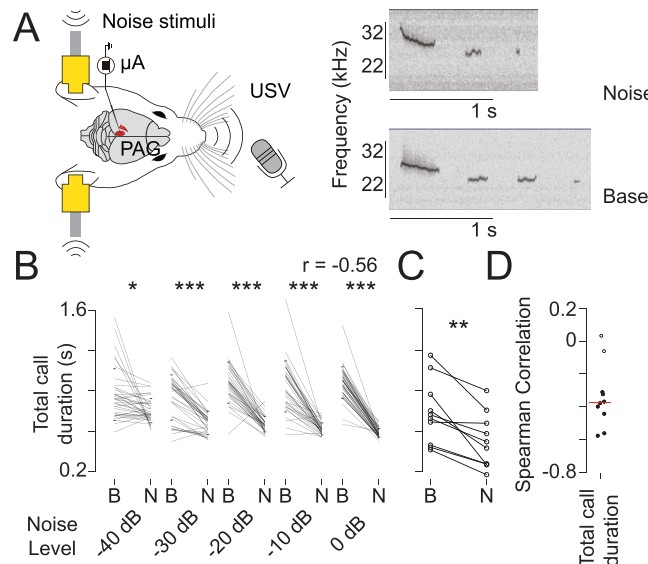

**Fig. 4 | White noise induces a reduction in total call duration. A** Experiment design. Left panel: Anesthetized rats produce sequences of calls after PAG stimulation, either during white noise stimulation or without noise stimulation. Vocalizations are recorded while noise stimuli are delivered through earphones. Right panels: traces of call during noise condition (up) or during baseline (bottom). During noise, rats produce fewer calls, which are shorter in total (Scale bars = 1 s). **B** Example rat showing a reduction in call duration during different noise levels. Each line represents the paired comparison between a noise condition and the subsequent baseline condition. Spearman Correlation between the decrease in total call duration and dB level is indicated in the upper right corner (B = Baseline, N = Noise; * = $p < 0.05$, *** = $p < 0.001$). **C** Rats consistently show a reduction in call duration. Each point represents a different subject in the within-subject comparison (** = $p < 0.01$). **D** The reduction in call duration is correlated with the increase in sound intensity. Each dot represents the Spearman (rank) correlation between noise intensity level (0, −10, −20, −30, −40 dB) and the reduction in call duration. Significant correlations are represented as filled circles ($p < 0.01$). The red line indicates the mean value.

"onset-suppressed cells," which exhibited a distinct cortical depth distribution. Although our method is limited in precisely identifying cortical layers, the observed profile suggests that these cells reside in the deeper layers of the rat auditory cortex. The most interesting observation is that these cells can anticipate critical vocalization-related parameters, such as total call duration and call occurrence, which have not been reported before. Onset suppressed cells are consistent with the findings of the authors, in regards to the decreased activity of these neurons during self-generated calls compared with call playback, nevertheless the extent of the suppression of onset suppressed cells is reduced since we only found a few cells showing inhibited response after onset. Also, previous work by Eliades and Wang demonstrated that suppression related to vocalization can begin even before vocal onset[18,19,21], which we did not observe in the pre-call activity of onset-suppressed cells (Fig. 1E). One possible reason for these discrepancies might be that rats and marmosets employ different physiological strategies for vocal production. Alternatively, the discrepancies may arise from the differences in the experimental paradigms. We propose that the predictive discharge from onset suppression cells may originate from internal brain signals arising in vocal control centers or reflecting endogenous auditory cortex state, which could impart vocal properties to the auditory cortex.

### Pharmacological intervention

It is well known that the human auditory cortex is critically important for normal speech[22]. However, very few attempts have been made to investigate the causal role of the auditory cortex in modulating vocal production in nonhuman animals. For example, no study has ever investigated whether the auditory cortex is necessary for modulating specific parameters of

vocalizations in rats. What we found is that pharmacological manipulations of the auditory cortex can significantly change call properties. More specifically, global inactivation of the auditory cortex using the GABAA receptor agonist muscimol in our behavioral paradigm in rats resulted in a significant increase in total call duration, accompanied by a decrease in mean call frequency. Notably, the only sufficient animal experiment comes from the marmoset monkey data shown by Eliades and Tsunada[23]. The authors found that electrical microstimulation of the auditory cortex is sufficient to drive vocal control in marmosets. However, as they pointed out, the position of the stimulation electrode that produced vocal changes is limited in a very small area and likely in the non-primary auditory cortex. Thus, the question still remained whether the primary auditory cortex is involved in modulating vocal changes. By injecting the GABAA_A receptor antagonist gabazine primarily into the primary auditory cortex in rats, with small spread into the secondary auditory cortex, we demonstrate that global activation of auditory cortical neurons reduces total vocal duration. It is well established that glutamatergic neurons in the PAG are essential for call production, with local GABAergic neurons exerting tight control over them. Based on our muscimol and gabazine data, we reason that auditory cortex glutamatergic neurons may preferentially target local GABAergic neurons in the lPAG. When the auditory cortex is globally inactivated, this likely leads to a disinhibition of lPAG glutamatergic neurons, resulting in increased vocalizations. Conversely, global activation of the auditory cortex may reduce the activity of PAG glutamatergic neurons through feedforward inhibition, leading to a decrease in vocalizations. Taken together, these bidirectional pharmacological manipulations provide the first direct causal evidence supporting the hypothesis that the auditory cortex plays a critical role in modulating vocal production in rats.

### Vocal modifications in response to noise

In follow-up experiments, we tested whether stimulating the auditory cortex by applying natural external white noise to the ear would produce effects similar to those seen in the pharmacological experiments. In line with the gabazine data, we found that the application of white noise reduced the total call duration of rats compared to their vocalizations during baseline conditions (Fig. 4), suggesting that shorter vocalizations in the presence of noise may convey a simplified message to the listener while reducing the chances of detection by predators. Additionally, we observed that the total call duration decreased proportionally with increasing white noise levels (Fig. 4B), indicating that rats, like many other mammalian species, can exhibit noise-induced vocal modifications (NIVMs). In fact, there is mixed evidence regarding the impact of white noise on vocalization duration in human studies[24,25]. Furthermore, along with the change in total call duration, we observed that the mean call frequency of rats increased proportionally with rising white noise levels, accompanied by moderate but significant increases in call intensity (Supplementary Fig. 6). Compared to the pharmacological data, the additional changes in vocal parameters observed with white noise application may be due to the fact that noise stimulation activates not only the auditory cortex but also other parts of the auditory system, which could contribute to NIVMs, such as changes in call frequency and intensity. Additionally, by precisely timing noise delivery around call onset, we found that noise presented before call onset increased call duration, whereas noise delivered during the call shortened call duration and overrode the pre-call effect. These findings suggest the presence of distinct mechanisms underlying pre-call and during-call (feedback) effects of noise on call duration and frequency. Taken together, our results suggest that rats may employ a common strategy by modifying different vocalization parameters—vocal duration, frequency, and amplitude- to affect the signal-to-noise ratio of acoustic signals in high levels of background noise.

### Future perspectives

In conclusion, our study significantly expands our understanding of the functional spectrum of the auditory cortex beyond mere sound processing and indicates a higher level of complexity in rodent vocalizations than previously demonstrated. Previous study has suggested that other cortical

areas, such as the posterior prelimbic cortex and cingulate area 2, are strongly involved in vocal modulation in rats[26], while our findings provide the first evidence that the rodent sensory region auditory cortex may contribute to the regulation of vocal production, opening new avenues for investigating how cortical circuits enable the flexibility in vocalization that is fundamental to human speech. Interestingly, neurons in the rodent medial prefrontal cortex receive direct inputs from the primary auditory cortex[27]. Thus, an alternative interpretation of the observed effect is that it may stem from modulation of an indirect pathway from auditory cortex to medial prefrontal cortex. Additional research is required to thoroughly examine the specific functional roles of different efferent pathways from the auditory cortex in modulating vocal production. It should be noted that, due to technical difficulties, we were unable to record changes in auditory cortex neural activity during gabazine infusion. It remains an intriguing question whether gabazine infusion might produce neural-activity changes in the auditory cortex similar to those evoked by white-noise stimulation. Furthermore, since our experiments were conducted on anesthetized rats, they did not provide a naturalistic setting, particularly for vocalization experiments during white noise stimulation. Although challenging, future studies are needed to examine whether rats exhibit similar NIVMs in a freely moving condition. Finally, manipulations of the auditory cortex limited to the time window of rat vocalizations could help elucidate the feedback mechanisms engaged in call production.

## Methods

### Animals
All studies were conducted in accordance with German guidelines on animal welfare and authorized by state office for health and social affairs (LAGeSo) (Permit No. G0095-21). We have complied with all relevant ethical regulations for animal use. All animals used were male Long-Evans rats provided by Janvier Labs, France. Rats were maintained in cages with food and water available ad libitum and 12 h light/12 h reversed dark cycles with controlled temperature and humidity.

### Surgery
For acute experiments in Figs. 1–4, rats were anesthetized with an intraperitoneal dose of urethane (1.5 × g/kg body weight). The hair overlying the rat's skull was shaved. Subsequently, the animal's head was fixed, utilizing ear bars and a stereotaxic frame, within a standard stereotaxic surgical apparatus (Narishige). The rat's body temperature was kept at 36 °C ± 0.5 °C throughout the surgery by utilizing a rectal temperature probe and a heating pad (FHC, Bowdoinham, Me, USA). An eye ointment (Corneregel ®; Bausch & Lomb) was applied over the eyes to prevent drying. Before starting the incision, 2% lidocaine was applied below the scalp as local anesthesia. Following that, the scalp was incised along the midline to expose the skull, and the bregma was labeled. The craniotomy was performed, and four skull-screws were placed in the surface of the skull and a head post was subsequently attached to the screws by employing a layer of UV-curable adhesive (Optibond; Altschul Dental, Mainz, Germany) and dental cement (Heraeus). For the experiments in Figs. 1–4, an ~1.5 × 1.5 mm sized craniotomy was performed above the left PAG. We then carefully inserted a stimulation electrode into the lPAG (AP − 6.5–7.0 mm from bregma; ML 0.75–0.8 mm; DV 4–4.5 mm). In order to evoke calls in the animal, the stimulation protocol of 0.2-s-long stimulation trains (100 Hz, 0.5 ms pulse duration) was used. In Figs. 1 and 2, to perform Neuropixels recordings, a second craniotomy was made above the left auditory cortex (stereotaxic coordinates: AP: 4 mm, ML: 7.3 mm). Subsequently, a 960-site, 384-channel Neuropixels probe (Phase 3B, IMEC) coated with DiO, DiI, or DiD was inserted into the auditory cortex with a ~30-degree angle.

### Spike sorting and assignment of units to histologically identified subdivisions
Spikes were detected from the high-pass filtered data using Kilosort 2.0[28], and then the output clusters were manually adjusted using the Phy GUI function (https://github.com/cortex-lab/phy). We used this version of

Kilosort because we didn't observe any sign of drifting, and the electrical stimulation created distortions in the drift correction methods introduced in later versions. Spikes occurring during stimulation were not included, and clusters were assessed qualitatively by their waveform, autocorrelogram, and cluster principal components (as described on our previous report[12]. Once units were sorted, they were assigned to the channel that showed the largest waveform amplitude. Probes were visually superimposed on the slices after shrinkage correction, and then channels were located matching the fluorescent track. Cortical thickness was estimated by averaging auditory cortex thickness from slices containing the probe, and then each neuron depth was estimated according to their channel number along the probe. For each cell, the assigned depth was proportional to its channel position in the probe, where a depth of 0 mm was assigned to the channel located in the brain surface, and a depth equals to the cortical thickness to the channel located in the lower limit of the cortex.

To validate our estimates, we performed current source density analysis (CSDA)[29] on our recordings whenever feasible. In two experiments, the resulting CSDA profiles aligned well with our Neuropixels–histology–based estimates.

### Procedure of local drug injection
The whole experiment was composed of 3 sessions: pre-, saline- and muscimol-treated sessions. In the pre-session, 0.2-s-long electrical stimulation trains (100 Hz, 100 pulses) were applied to the lPAG to evoke vocalizations, with no intervention in the auditory cortex of the rat. Saline was injected into the auditory cortex after the pre-session. The coordinates used for the auditory cortex injection in the rats were (three injection sites for each side): anterior– posterior, 4 or 5.3 mm; lateral–medial, 6.4 mm; dorsal–ventral, 1.5 mm, 2 mm, 2.5 mm. A glass pipette (10 µl; Drummond) featuring a delicate pulled tip (opening diameter of 8–18 µm, taper length approximately 8 mm), filled with saline, was carefully inserted into the auditory cortex of the brain. Afterwards, 0.3 µl of saline was infused for each site at a gradual flow rate (~0.07 µl per minute) utilizing an oil hydraulic micromanipulator system (MO-10, Narishige, Japan) linked to the plunger of the injection pipette, as described previously[30]. Following the injection, the micropipette remained stationary for a few minutes to facilitate the diffusion of the saline. Subsequently, electrical stimulation was conducted in the lPAG using identical parameters to elicit vocalizations. Following the saline session, to inactivate the auditory cortex during vocalization, the GABA_A receptor agonist muscimol (2 µg/µL in saline, G019, Sigma) mixed with Alexa Fluor-conjugated cholera toxin subunit B (CTB–Alexa 555, or CTB–Alexa 647; Thermo Fisher Scientific) was introduced into the auditory cortex, employing the same procedures described above.

For the auditory cortex activation experiment, the GABA_A receptor antagonist gabazine (500 µM; SR95531, Tocris) was mixed with Alexa Fluor-conjugated cholera toxin subunit B (CTB–Alexa 555, or CTB–Alexa 647; Thermo Fisher Scientific) during the preparation. The other procedures were similar to the protocol used for the muscimol experiment as described above.

### Acoustic stimuli
White noise stimuli were administered through an MF1-S 1 Multi Field Speaker (Tucker-Davis Technologies, Inc., Alachua, USA), positioned inside the ears of the rat. The noise stimulus was designed in Matlab and comprised all frequencies between 1 Hz and 100 kHz in equal amplitude. During the experiment, auditory stimuli were recorded and monitored by an Avisoft condenser microphone (CM16/CMPA-5 V, frequency range 10–200 kHz, Avisoft Bioacoustics) situated 15 cm in front of the rat. Data were collected at a sampling frequency of 250 kHz and 16-bit resolution using Avisoft-RECORDER USGH software (Avisoft Bioacoustics). Calls were detected using DeepSqueak version 3 (https://github.com/DrCoffey/DeepSqueak). Playback stimuli were administered immediately after recording 5 min of animal calling, through an ultrasonic dynamic Vifa speaker (Avisoft Bioacoustics), positioned 10 cm away from the head of the rat. The recorded

calls for playback had an equivalent frequency and duration distribution to the emitted calls from the animal.

## Experimental design of noise stimulation

Anesthetized rats were stimulated in the lPAG, which produced sequences of low-frequency calls. The current for the electrical stimulation was kept constant for each individual animal during the entire experiment. Sequences with (Baseline) and with white noise (Noise) stimulus were intercalated. Noise started 5 s before lPAG stimulation, and stopped 2 s after (call sequences did not last more than 2 s). For Baseline sequences, a period of 5 s without calls or noise preceded the lPAG stimulation. Each experiment consisted of 5 Noise conditions, were sequences of lPAG stimulation were repeated during ~5 min. The 5 conditions were obtained by adjusting the max white noise intensity to 75 dB (0 dB), and then reducing it digitally using the Avisoft interface by −10dB, −20dB, −30dB, and −40dB. The order of the conditions was randomized in each experiment. Call playbacks were delivered before the beginning of the experiment, were 5 min of calls were evoked, recorded, and then played back to the animal. When different playback intensities were presented, the playback blocks were also randomized.

## Histology

After all the experiments, the rats were administered a lethal dose of urethane and underwent transcardial perfusion with a flush of prefixative solution (0.9% NaCl, 0.02 m phosphate buffer) followed by a 4% paraformaldehyde solution (PFA). Subsequently, the brains of the rats were extracted from the skull and postfixed in a 4% paraformaldehyde (PFA) solution overnight at 4 °C. The brains were embedded in 4% agarose and sectioned into 80-μm-thick coronal slices with a Vibratome (Mikrom, HM 650 V, ThermoFisher Scientific). The coronal sections were mounted directly on slides using Fluoromount mounting medium (Biozol).

For marking the DiI-, DiD- or DiO-labeled electrode tracks in Neuropixels recordings, fluorescent pictures were photographed using a Leica DM5500 fluorescence microscope (Leica Microsystems). For identifying the muscimol or gabazine injection site, images of CTB–Alexa Fluor 555 or CTB–Alexa Fluor 647 were also taken on the Leica DM5500 fluorescence microscope (Leica Microsystems). The recording and injection sites were generally discernible across several consecutive slices and manually matched to a rat brain atlas[31]. If the primary auditory cortex was not successfully targeted, the corresponding data were excluded from analysis. For each brain, we considered a volume shrinkage of 20% of our fixed tissue during the alignment process.

## Data analysis

For all subsequent analysis we used MATLAB (the MathWorks Inc., 2023b) and GraphPad Prism 8 software.

## Vocalization analysis

For each call, principal frequency, duration, and amplitude were estimated using DeepSqueak version 3[32]. Total call duration was defined as the added durations of each individual call during a call series. Mean call frequency was defined as the mean principal frequency of each individual call in a series. Then, for each Baseline and Noise sequence, the total call duration and the mean call frequency were estimated. Paired T-test were used for assessing difference between consecutive conditions (Fig. 4 and Supplementary Fig. 6). We also pooled together all Noise and Baseline sequences, for each animal, and estimated the mean total call duration (or mean principal frequency) between these two pooled conditions, and assessed the within difference using also a paired T-test (Fig. 4C). To estimate the correlation between duration reduction (or principal frequency increment) and noise level, we first estimated the difference between the total call duration (or mean principal frequency) of each baseline call sequence and the total call duration of the preceding noise sequence; we then calculated, for each animal, the Spearman (rank) correlation between these values and their

corresponding noise levels (0, −10, −20, −30, and −40 dB, Fig. 4B upper-right corner).

We also compared call features at the level of individual calls by taking advantage of the stereotyped structure of the call sequences. In most cases, after electrical stimulation, the first elicited call was the longest, with each subsequent call becoming progressively shorter; sequences typically ended after 2–5 calls. Occasionally, the first call was split into a short call followed by a long one, after which the sequence proceeded normally. Under noise conditions, the overall sequence structure was preserved, although the first call was usually already significantly shorter than during baseline.

To compare individual calls across conditions, we selected pairs of noise and subsequent baseline sequences that shared the same structure—that is, a long first call followed by calls of decreasing duration. We then ranked the calls according to their position in the sequence and performed paired comparisons between calls of the same rank. Because baseline sequences generally contained more calls, we restricted the analysis to the number of calls present in the shorter sequence. Similarly, when comparing variables like total call duration, we used only as many baseline calls as were present in the preceding noise sequence.

## Call and playback responses

Call and playback analyses were performed in all four animals. Playback at different decibel levels was conducted in one of them ($n = 56$ neurons). To prevent any effect of stimulation on the neural responses, for this and further analysis, we selected only calls starting at least 200 ms after the end of the electrical stimulation. Then we estimate call and playback responses by aligning neural activity to call and playback onset and offset. For each condition, both offset and onset activity were normalized (z-scored) using the 200 ms window before call onset. Call-Playback difference was estimated by subtracting the normalized rates. We used the exact same calls to compare call and playback activity.

## Neural response classification

In general, we first focused on detecting cells with any onset or offset response, followed by classifying these cells into different cell types. Pre-call cells were identified using a slightly different approach, as they were detected among cells that did not exhibit any onset or offset response.

The detection procedure was based on the response of cells to call onset and offset. We focused on the increase or decrease in neuronal activity within a 50 ms window following call onset and a 50 ms window preceding call offset. We normalized the rate changes to baseline activity within the 200 ms window prior to call onset, excluding any calls that occurred less than 200 ms after the end of electrical stimulation. For the detection procedure, we set a threshold of 1.96 times the standard deviation of these values.

Our detecting procedure aimed to identify cells with any onset or offset response by iteratively defining a population of non-responsive cells (Supplementary Fig. 1A). At the start of the detection process, all cells were considered non-responsive. In each iteration, threshold values were estimated based on the entire population of non-responsive cells. For the first iteration, we calculated the standard deviation of all cells' responses to set the threshold. If a cell exceeded any of the defined thresholds, it was classified as responsive. The remaining cells were considered non-responsive, and the threshold value was recalculated. This process continued until no more responsive cells were detected.

We then examined the PSTH of all classified cells, paying special attention to cells whose responses were close to the threshold or had low firing rates (the response index may inaccurately classify cells as responsive or non-responsive if their firing rate is too low). If needed, cells were reclassified based on their PSTH. This adjustment can be observed in the green points within the non-responsive region and the gray dots or squares outside it (squares indicate pre-call cells initially classified as non-responsive).

Once we had defined our responsive and non-responsive populations, we proceeded to classify cells by response type (Supplementary Fig. 1B).

From the population of responsive cells, we first classified those exhibiting any onset response as onset cells, further categorizing them as activated or suppressed based on their call-playback onset difference. The remaining responsive cells were classified as ramping cells, and we further divided them into activated or suppressed based on their call-playback offset difference. To detect pre-call neurons, we analyzed PSTHs of all neurons and classified cells as pre-call cells if they showed a clear activity peak before call onset in their PSTH. This approach was used due to the typically low firing rate of these cells, which made the threshold method less effective for classification. The results of both the detection and classification procedures are summarized in Supplementary Fig. 1C, D.

### Depth comparison
For a given cell, we defined its eccentricity as the absolute distance of the neuron's depth to the median depth of the whole population of auditory cortex neurons. Then, for a given functional cell type, depth and eccentricity distributions were compared with the distribution of the remaining auditory cortex cells. Depth and eccentricity between these populations were compared using a t-test and or a signed rank test depending of the type of distribution of the data.

### Call duration correlations and prediction of call occurrence using SVM
For call duration correlations, only calls occurring at least 100 ms after the end of stimulation were selected. This procedure allowed us to increase the number of calls for analysis while keeping the analysis window free of stimulation artifacts. In order to pool different sessions together, call lengths and population rates were z-score within each session. To estimate pooled linear models, we performed a mixed linear model analysis (*lme* Matlab function) to account for the effect of each animal. This analysis prevents the linear correlations is explained for a strong correlation in only one dataset.

For the call occurrence analysis, we used Matlab 2023 classification toolbox. In order to predict the termination of a call sequence, we analyzed calls that had an inter-call interval with respect to the following call longer than 250 ms. (Supplementary Fig. 2). This allowed us to define a call-free predicting interval. We further sub-select from these calls two comparable sets of baseline and noise calls that had equal size and equivalent call duration distributions (defined by a Kolmogorov test). This procedure resulted in a set of calls of equal call duration distribution of baseline and noise calls that were not followed by a call in the next 250 ms after call offset. We used then the rate during the predicting interval after the "current call" to train a support vector machine model (SVM) to predict if another call is coming or not. We used a Gaussian kernel and a 90 and 10% of the calls for training and testing the model, respectively.

### Statistics and reproducibility
A total of 4 male rats were used for neural activity measurements, and 11 male rats were used for behavioral experiments. Data were processed using standard machine learning and statistical methods implemented in MATLAB, including mixed-effects linear models, support vector machines, t-tests, and nonparametric tests. An overview of the analytical approach is provided below, and the code used to generate all figures is publicly available at: https://github.com/miguelconcham/Auditory-Cortex-Modulates-Call-Duration-in-Rats.

A total of 19 male rats were used for the pharmacological intervention. Analysis of the pharmacological experiments was performed by using GraphPad Prism 8 software. The data were expressed as mean ± SD. For comparisons in Fig. 3 and Supplementary Fig. 5, one-way ANOVA with Tukey's multiple-comparisons test was used.

### Reporting summary
Further information on research design is available in the Nature Portfolio Reporting Summary linked to this article.

## Data availability
The source data for Fig. 3 and Supplementary Fig. 5 are provided with this paper (Supplementary data 1). Source data for all other figures, including animal behavioral data and neural recordings, together with the corresponding MATLAB code, are available at https://github.com/miguelconcham/Auditory-Cortex-Modulates-Call-Duration-in-Rats.

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

## Acknowledgements

We thank Undine Schneeweiß and Tanja Wölk for technical support. This work was supported by the BCCN, Humboldt-Universität zu Berlin, and the Deutsche Forschungsgemeinschaft (DFG, German Research Foundation) under Germany's Excellence Strategy—EXC-2049—390688087 (NeuroCure), Deutsche Forschungsgemeinschaft (DFG, German Research Foundation)—Project number 532521431 (Research Unit FOR 5768: Neural basis of vocal communication), the ERC Synergy Grant 'BrainPlay - the self-teaching brain', and the DFG—project number 327654276 SFB 1315. We acknowledge support from the Open Access Publication Fund of Humboldt-Universität zu Berlin.

## Author contributions

W.T., M.C., and M.B. conceived and designed the study. W.T. and M.C. performed the experiments. W.T. and M.C. analyzed the data. M.B. provided technical support and resources. W.T., M.C., and M.B. drafted the manuscript. All authors contributed to data interpretation, critically revised the manuscript, and approved the final version.

## Funding

## Competing interests

The authors declare no competing interests.
