## [Transparent Peer Review file · Communications Biology]

Auditory Cortex Modulates Call Duration in Rats

Corresponding Author: Dr Michael Brecht

Version 0:

Reviewer comments:

Reviewer #1

(Remarks to the Author)

The manuscript by Tang et al. investigates an important question of whether the auditory cortex can modulate vocal output. The authors compared neural activity in auditory cortex during calls evoked by PAG stimulation and during playback of recorded calls in anesthetized rats. They found cortical neurons that showed different activity during vocalization vs. playback, suggesting a role in vocal production. They further performed causal experiments, showing that muscimol or gabazine infusion into the auditory cortex differentially modulate call duration and frequency. These results demonstrate that auditory cortical activity can alter vocal output. Finally, the authors show that background noise also influences call duration and other vocal parameters. While this effect may not be specific to auditory cortex, the results nonetheless support the idea that auditory input can shape vocal output.

Overall, the study convincingly demonstrates that changes in auditory cortical activity can modulate evoked calls in rats. Despite limitations such as inherently poor temporal precision of pharmacological manipulations and the use of induced vocalizations under anesthesia, the results are interesting, particularly in light of prior work showing that mouse USVs are unaffected by deafening. The findings are consistent with studies in marmosets (e.g. Eliades and Tsunada, 2018), which reported cortical suppression during vocalization and showed that cortical microstimulation can influence vocal output. By providing evidence that the auditory cortex not only monitors but also modulates vocal output, this study adds to the growing body of evidence for the involvement of auditory pathway in motor function.

Specific comments:

1. The authors emphasize “direct” control of vocal output by the auditory cortex, implying that the observed effects arise from direct projections from auditory cortex to vocal production areas, as proposed in the discussion. While this is a plausible interpretation, alternative pathways are also possible, and the current data do not identify a “direct” control circuit. The authors should temper this interpretation and discuss alternative possibilities.
2. Fig 1E-H: It seems that the emitted calls used to align neural activity varied substantially in duration, frequency, and intensity. In comparing neural activity during calls and playback, it is unclear how well the playback repertoire matched the emitted calls, or how playback intensity was calibrated relative to emitted calls. If evoked calls are heterogeneous, I am not sure how meaningful it is to “average” across them. Likewise, if the playback calls were not well matched to the evoked calls, I am not sure how meaningful it is to compare. Please clarify.
3. To illustrate how the averaged firing rate heatmaps in Fig 1E-G were derived, I recommend moving some of the raster plots shown in supplementary Fig 1 into the main figure.
4. Fig 3 vs Fig 4: In the examples shown in Fig 3B, the first call segment appear longer in muscimol but not the sequence since the second call segment is lacking, whereas in Fig 4A, “duration” seems to be measured across an entire sequence (3-4 call segments). Please clarify whether “total call duration” refers to the duration of an entire sequence evoked by a stimulation or individual call segments.
5. In Fig 4, call duration decreases during white noise stimulation, similar to gabazine infusion. The authors seem to interpret this as the effect of increased cortical activity (e.g., “In line width gabazine data, we found that ... white noise reduced ... “ or “... noise stimulation activates not only the auditory cortex ...”). However, during prolonged while noise, overall cortical activity may not be higher than spontaneous activity. During gabazine infusion, neural activity is likely to increase but may

do so in highly abnormal patterns. Since neural activity was not recorded during either gabazine or noise stimulation, the nature of activity changes remains uncertain. This point should be discussed more carefully.

6. Supplementary Fig 5: While the example fluorescent dye injections appear well localized to A1, the dyes mixed with the dyes (rather than conjugated to a fluorophore) may have diffused more widely, potentially beyond A1. This possibility should be acknowledged.

7. In the abstract, the phrase “flexible vocal behavior” is not immediately clear. I suggest rephrasing or being more specific to help readers better appreciate the motivation and significance of the study.

Reviewer #2

(Remarks to the Author)

Summary

The manuscript Auditory Cortex Modulates Call Duration in Rats by Wei Tang, Miguel Concha-Miranda and Michael Brecht investigates the putative role of the auditory cortex, a primarily sensory area, in the modulation of vocal production in rats. The rationale for the study is sound, as this question is still unresolved in particular for rodents. The authors describe three complementary sets of experiment aimed at 1) characterizing the neuronal responses in the auditory cortex to self-emitted vocalizations, 2) characterize the animal's vocalizations during pharmacological modulation of the activity of the auditory cortex and 3) characterize these vocalizations during sound-evoked modulation of the activity of the auditory cortex. These results are important and demonstrate a role for the auditory cortex in the modulation of vocal production. The manuscript is overall well written, although several points need to be addressed to improve it.

Major comments

1) Vocalization analysis. Different parameters of the vocalizations emitted in response to a IPAG electrical stimulations are quantified. These parameters are different between the different experiments. In first experiment, the authors quantify the call duration and occurrence, in the second the total call duration and mean call frequency and in the third the total call duration, call frequency and vocal intensity. The influence of auditory cortex activity would be clearer if the same parameters were reported in all three analyses. In addition, the author could quantify the duration of individual calls, the frequency modulation of the calls and their variability to provide insight on the nature of the modulation exerted by the auditory cortex. Previous study showed that Gabazine increases the reliability and uniformity of sensory evoked responses in the auditory cortex. In consequence, in the Gabazine condition it is possible that the modulation exerted by the auditory cortex on vocal production become less variable, leading to more stereotypical calls.

2) Gabazine application can trigger epileptiform activity in neuronal networks by shifting the excitatory / inhibitory balance toward excitation. How did the author validate that the concentration used in their protocol did not trigger such activities?

3) Speaker and microphone calibration: for playback experiments, calibration of the speaker and of the microphone is crucial, to avoid a difference in neuronal response caused by a spectral mismatch between the emitted call and the playback. Could the authors provide details on the calibration procedure used?

4) Depth analysis of diagonally inserted probes is unreliable. To have a more accurate estimation of the anatomical location of the recording sites, the authors could apply a current source density analysis on the local field potential of their recordings (Pettersen et al 2006). This would strengthen the claim that pre-call neurons are found in superficial layers and onset-activated and -suppressed neurons in deeper layers. See: Pettersen KH, Devor A, Ulbert I, Dale AM, Einevoll GT. 2006. Current-source density estimation based on inversion of electrostatic forward solution: effects of finite extent of neuronal activity and conductivity discontinuities. *Journal of Neuroscience Methods* 154:116–133

Minor comments

1) Could the author confirm that neuronal recordings and pharmacological manipulations target the primary auditory cortex, and how the proper electrode localization is controlled? This is not clearly stated in the method section.

2) Supplementary figure 2: For these experiments (playback at different intensities), no N are reported (number of rats, total number of neurons recorded)

3) Fig Suppl 1C: The color for the non responsive region is not visible in the legend.

4) Fig 1E: There is a mismatch between the number of neurons reported in the figure (109) and in the text (103).

5) Figure 2 and supplementary 2 and 3: In the legend, please indicate what is the red shade in the raster plots.

6) References: Could all authors be referred to in the reference section (avoiding “et al”)

7) 140: “To test this possibility, we measure the response latency to both the animal own calls and playback stimuli, by calculating the time between call or playback onset and the first detected spike”.

The results of this analysis are currently not presented in the paper or figures.

8) 165: “we recorded multiple onset suppressed cells in some of our experiments”

Please provide the number of neurons in these populations

9) 216: “No vocalization changes were observed, however (data not shown).”

This result is important, as it suggest that the modulation of vocal production needs a change of excitability of the entire cortical map. If space permit, the author should consider showing these data.

10) 266: “We also performed an equivalent analysis comparing each call, and we found similar effects.”

Could the author provide more details on this additional analysis?

11) 294: “Wilcoxon”

12) 583: Identification of responsive cells. Could the author elaborate on the advantage of using an iterative threshold value compared to a fixed one?

Version 1:

Reviewer comments:

Reviewer #1

(Remarks to the Author)

The authors have addressed all of my concerns. I have no further comments.

Reviewer #2

(Remarks to the Author)

I thank the authors for the careful consideration of my comments, and for generating additional explanatory figures. The authors have answered all of my concerns, and I have no further issue to raise with this manuscript.

Dear Referees and Editors,

We are thankful for the fair and constructive criticism of our article. In response to these criticisms, we performed additional experiment and analysis, added new reviewer figures and thoroughly revised our paper. The following is summary of what we have done:

- We report vocalization variability during GABAZINE application (Reviewer figure 1)**
- We performed current source density analysis according to Pettersen et al 2006 (Reviewer figure 2)**
- We report the effects of calls elicited by electrical stimulation at different depths within the auditory cortex (Reviewer figure3).**
- We performed additional experiment to test if the GABAZINE injection trigger epileptiform activity in neuronal networks.**
- We implemented all referee suggestions as detailed below pint-by-point.**

We think these changes greatly improved the ms. We repeat editorial and referee concerns in black print before responding to them in red.

Reviewer #1:

The manuscript by Tang et al. investigates an important question of whether the auditory cortex can modulate vocal output. The authors compared neural activity in auditory cortex during calls evoked by PAG stimulation and during playback of recorded calls in anesthetized rats. They found cortical neurons that showed different activity during vocalization vs. playback, suggesting a role in vocal production. They further performed causal experiments, showing that muscimol or gabazine infusion into the auditory cortex differentially modulate call duration and frequency. These results demonstrate that auditory cortical activity can alter vocal output. Finally, the authors show that background noise also influences call duration and other vocal parameters. While this effect may not be specific to auditory cortex, the results nonetheless support the idea that auditory input can shape vocal output.

Overall, the study convincingly demonstrates that changes in auditory cortical activity can modulate evoked calls in rats. Despite limitations such as inherently poor temporal precision of pharmacological manipulations and the use of induced vocalizations under anesthesia, the results are interesting, particularly in light of prior work showing that mouse USVs are unaffected by deafening. The findings are consistent with studies in marmosets (e.g. Eliades and Tsunada, 2018), which reported cortical suppression during vocalization and showed that cortical microstimulation can influence vocal output. By providing evidence that the auditory cortex not only monitors but also modulates vocal output, this study adds to the growing body of evidence for the involvement of auditory pathway in motor function.

Specific comments:

1. The authors emphasize ‘direct’ control of vocal output by the auditory cortex, implying that the observed effects arise from direct projections from auditory cortex to vocal production areas, as proposed in the discussion. While this is a plausible interpretation, alternative pathways are also possible, and the current data do not identify a ‘direct’ control circuit. The authors should temper this interpretation and discuss alternative possibilities.

Comment: We agree.

Change: We rewrote the manuscript and discussed alternative possibilities pathways in our discussion.

2. Fig 1E-H: It seems that the emitted calls used to align neural activity varied substantially in duration, frequency, and intensity. In comparing neural activity during calls and playback, it is unclear how well the playback repertoire matched the emitted

calls, or how playback intensity was calibrated relative to emitted calls. If evoked calls are heterogeneous, I am not sure how meaningful it is to ‘average’ across them. Likewise, if the playback calls were not well matched to the evoked calls, I am not sure how meaningful it is to compare. Please clarify.

Comment: The vocalizations show considerable variability in length and intensity, with some variation in frequency. For our playback experiments, we used the exact same sequence of vocalizations that was employed to obtain the call responses (as illustrated in Supplementary Figure 2), ensuring identical lengths and frequencies. The playback intensity was adjusted to match the level of the recorded vocalizations (see details in following comments).

Change: None.

3. To illustrate how the averaged firing rate heatmaps in Fig 1E-G were derived, I recommend moving some of the raster plots shown in supplementary Fig 1 into the main figure.

Comment: We appreciate the suggestion, and we now modified Figure 1, to include some of the examples from supplementary Figure 2.

Change: We now included example raster plots and PSTHs in main Figure 1.

4. Fig 3 vs Fig 4: In the examples shown in Fig 3B, the first call segment appear longer in muscimol but not the sequence since the second call segment is lacking, whereas in Fig 4A, ‘duration’ seems to be measured across an entire sequence (3-4 call segments). Please clarify whether ‘total call duration’ refers to the duration of an entire sequence evoked by a stimulation or individual call segments.

Comment: The total call duration represents the sum of individual call durations withing a sequence evoked by a stimulation. This definition is described in the vocalization analysis section of the Methods.

Change: None.

5. In Fig 4, call duration decreases during white noise stimulation, similar to gabazine infusion. The authors seem to interpret this as the effect of increased cortical activity (e.g., ‘In line with gabazine data, we found that ‘white noise reduced ...’ or ‘noise stimulation activates not only the auditory cortex’). However, during prolonged white noise, overall cortical activity may not be higher than spontaneous activity. During gabazine infusion, neural activity is likely to increase but may do so in highly abnormal patterns. Since neural activity was not recorded during either gabazine or noise stimulation, the nature of activity changes remains uncertain. This point should be discussed more carefully.

Comment: We agree.

Change: We rewrote the manuscript discussion and carefully clarified the interpretation of our gabazine results, particularly in relation to the noise stimulation experiments.

Supplementary Fig 5: While the example fluorescent dye injections appear well localized to A1, the drugs mixed with the dyes (rather than conjugated to a fluorophore) may have diffused more widely, potentially beyond A1. This possibility should be acknowledged.

Comment: We agree.

Change: In the discussion part, we have acknowledged this possibility by stating: "By injecting the GABAA_A receptor antagonist gabazine primarily into the primary auditory cortex in rats, with small spread into the secondary auditory cortex, we demonstrate that global activation of auditory cortical neurons reduces total vocal duration."

7. In the abstract, the phrase "flexible vocal behavior" is not immediately clear. I suggest rephrasing or being more specific to help readers better appreciate the motivation and significance of the study.

Comment: We agree.

Change: In the revised manuscript, we rephrased the first sentence of the abstract to make our expression clearer.

Reviewer #2:

Summary

The manuscript Auditory Cortex Modulates Call Duration in Rats by Wei Tang, Miguel Concha-Miranda and Michael Brecht investigates the putative role of the auditory cortex, a primarily sensory area, in the modulation of vocal production in rats. The rationale for the study is sound, as this question is still unresolved in particular for rodents. The authors describe three complementary sets of experiment aimed at 1) characterizing the neuronal responses in the auditory cortex to self-emitted vocalizations, 2) characterize the animal's vocalizations during pharmacological modulation of the activity of the auditory cortex and 3) characterize these vocalizations during sound-evoked modulation of the activity of the auditory cortex. These results are important and demonstrate a role for the auditory cortex in the modulation of vocal production. The manuscript is overall well written, although several points need to be addressed to improve it.

Major comments

1) Vocalization analysis. Different parameters of the vocalizations emitted in response to a IPAG electrical stimulations are quantified. These parameters are different between the different experiments. In first experiment, the authors quantify the call duration and occurrence, in the second the total call duration and mean call frequency and in the third the total call duration, call frequency and vocal intensity. The influence of auditory cortex activity would be clearer if the same parameters were reported in all three analyses. In addition, the author could quantify the duration of individual calls, the frequency modulation of the calls and their variability to provide insight on the nature of the modulation exerted by the auditory cortex. Previous study showed that Gabazine increases the reliability and uniformity of sensory evoked responses in the auditory cortex. In consequence, in the Gabazine condition it is possible that the modulation exerted by the auditory cortex on vocal production become less variable, leading to more stereotypical calls.

Comment: We had planned to report all three parameters in our analyses. However, only call duration and call occurrence emerged as primary phenotypes in our manuscript; therefore, we present these data in the main figures. In contrast, because we did not detect robust neural responses for either frequency modulation or vocal intensity, the corresponding pharmacological and behavioral manipulations are included in the supplementary figures. To assess whether Gabazine alters the variability of call production, we examined the standard deviation (SD) of call duration, frequency, and amplitude across the pre-injection, saline, and Gabazine conditions. We did not observe any significant Gabazine-induced changes in the SD of these parameters. Please see the Reviewer figure 1 below.

Reviewer figure 1

Change: None.

2) Gabazine application can trigger epileptiform activity in neuronal networks by shifting the excitatory / inhibitory balance toward excitation. How did the author validate that the concentration used in their protocol did not trigger such activities?

Comment: The referee raises an important point.

Change: In order to answer this question, we made a strong effort to perform recordings during the Gabazine experiments, and performed 6 repeated experiments. Although Gabazine again led to a reduction in total call duration, the Neuropixel recordings were compromised by significant drift when the injection pipette penetrated the auditory cortex and by significant electrical noise, making it impossible to continue tracking the recorded neurons. Therefore, we were not able to determine whether the concentration of Gabazine employed in our experiments might have elicited such activities.

3) Speaker and microphone calibration: for playback experiments, calibration of the speaker and of the microphone is crucial, to avoid a difference in neuronal response caused by a spectral mismatch between the emitted call and the playback. Could the authors provide details on the calibration procedure used?

Comment: To calibrate dB output of speakers we followed manufacturer instruction (<https://www.avisoft.com/Help/SASLab/calibration.htm>). We then, equated the dB level of the playback to the one of the recorded vocalizations. Here it is worth mentioning a limitation of our approach. Even if we match the exact dB value of the emitted vocalization, the actual dB level of the evoked vocalization that reaches the cochlea is difficult to measure in practice—especially considering that the animal's own call may stimulate the cochlea through bone conduction. Nevertheless, we assume that after fixing playback to the call's intensity, the remaining differences in cochlear sound intensity between self-vocalization and playback—arising from different source locations and potential differences in bone conduction—are likely of

smaller magnitude than the differences examined during playback at different intensities (up to 30 dB).

Since the call-versus-playback effects were qualitatively different from the changes observed across playback intensities (for example, Supplementary Figure 2F), we believe that the call–playback differences are not expected to be explained solely by intensity differences.

Change: None.

4) Depth analysis of diagonally inserted probes is unreliable. To have a more accurate estimation of the anatomical location of the recording sites, the authors could apply a current source density analysis on the local field potential of their recordings (Pettersen et al 2006). This would strengthen the claim that pre-call neurons are found in superficial layers and onset-activated and -suppressed neurons in deeper layers. See: Pettersen KH, Devor A, Ulbert I, Dale AM, Einevoll GT. 2006. Current-source density estimation based on inversion of electrostatic forward solution: effects of finite extent of neuronal activity and conductivity discontinuities. *Journal of Neuroscience Methods* 154:116–133

Comment: We agree that it is possible to estimate the functional extent of cortical layers using NPX probes through current source density analysis (CSDA). Nevertheless, under our experimental paradigm, this approach has important limitations. First, our auditory stimuli consist of each animal's own recorded calls. These vocalizations are complex and show substantial variability both within and across animals. Such variability makes our CSDA results difficult to compare with those reported in previous studies (Aukstulewicz et al., 2023; Rimehaug et al., 2023; Szymanski et al., 2009), which in turn limits our ability to reliably identify cortical layers based on the CSDA profiles. In addition, because both the vocalizations and the exact probe position and angle varied across sessions, comparing CSDA profiles between our recordings is also challenging.

We performed CSDA following Pettersen et al. (2006) using the implementation available at <https://github.com/espenhgn/iCSD>. The parameters used were: $diam = 500E-6$ m, $h = 100E-6$ m, $\sigma = 0.3$ S/m, $\sigma_{top} = 0.3$ S/m. The results of the analysis, together with the recorded neurons aligned to our cortical estimates, are shown in the next figure.

Reviewer figure 2

The CSDA profiles varied substantially between experiments, with no consistent pattern across recordings. Moreover, in two of our experiments the CSDA resulted in uncommon sink and source patterns, likely due an unpredicted low frequency LFP artifact. Nevertheless, in two of our experiments the CSDA pattern resembled some previous reports (Reviewer figure 2). In these cases, the cortical borders estimated using CSDA partially coincided with our histology matching procedure, nevertheless we couldn't identify cortical layers.

We conclude that the two CSDA profiles showing a clearly identifiable cortical column are broadly consistent with our cortical assignments, requiring only minor adjustments in depth. The remaining two recordings appear inconclusive. Therefore, we chose to retain our original depth assignments, which are based solely on cortical thickness and probe length, while explicitly acknowledging these limitations in the manuscript.

Change: We discuss the limitation of our depth analysis, we now report the CSDA in the revised method, and added the suggested reference.

Minor comments

1) Could the author confirm that neuronal recordings and pharmacological manipulations target the primary auditory cortex, and how the proper electrode localization is controlled? This is not clearly stated in the method section.

Comment: The Neuropixels probe was coated with DiO, DiI, or DiD before each recording. For the pharmacological experiments, muscimol or gabazine was mixed with CTB–Alexa 555 and/or CTB–Alexa 647. After each experiment, histology was

performed, and the targeted area was compared with the brain atlas to verify accurate targeting.

Change: We have provided details on how we confirmed targeting of the primary auditory cortex after neuronal recordings and pharmacological manipulations in the revised method section.

2) Supplementary figure 2: For these experiments (playback at different intensities), no N are reported (number of rats, total number of neurons recorded)

Comment: We apologize for the missing information.

Change: We report now the missing information in methods, under “Call and Playback responses.”

3) Fig Suppl 1C: The color for the non responsive region is not visible in the legend.

Comment: We appreciate the observation.

Change: We now include it in the legend.

4) Fig 1E: There is a mismatch between the number of neurons reported in the figure (109) and in the text (103).

Comment: We appreciate the observation.

Change: We changed the figure, to match to correct number.

5) Figure 2 and supplementary 2 and 3: In the legend, please indicate what is the red shade in the rater plots.

Comment: We appreciate the observation.

Change: We added the missing information in the legends.

6) References: Could all authors be referred to in the reference section (avoiding ‘et al’)

Comment: Because of the corrupted (garbled) text in the question, we’re unable to understand it clearly.

Change: Regarding the references in the manuscript, we have used the standard Nature referencing style as recommended by *Communications Biology* (<https://www.nature.com/commsbio/submit/submission-guidelines#references>).

7) 140: 'To test this possibility, we measure the response latency to both the animal own calls and playback stimuli, by calculating the time between call or playback onset and the first detected spike';

The results of this analysis are currently not presented in the paper or figures.

Comment: We apologize for the confusion. This sentence was intended to introduce the upcoming analyses.

Change: We now clarify this more explicitly in the text.

8) 165: 'we recorded multiple onset suppressed cells in some of our experiments';

Please provide the number of neurons in these populations

Comment: we apologize for the missing information.

Change: we now report the number of onset suppressed cells for each recording.

9) 216: 'No vocalization changes were observed, however (data not shown).'; This result is important, as it suggest that the modulation of vocal production needs a change of excitability of the entire cortical map. If space permit, the author should consider showing these data.

Comment: As suggested by the reviewer, we have made a new figure (Reviewer figure 3) comparing mean frequency, mean power, and total call length between the baseline (Base) and A1-stimulation conditions across different depths of the primary auditory cortex. We are uncertain whether this figure should be included in the manuscript. If the reviewer considers it necessary, we will include it as a supplementary figure.

Reviewer figure 3

Change: None.

10) 266: We also performed an equivalent analysis comparing each call, and we found similar effects; Could the author provide more details on this additional analysis?

Comment: We also compared call features at the level of individual calls by taking advantage of the stereotyped structure of the call sequences. In most cases, after electrical stimulation the first elicited call was the longest, with each subsequent call becoming progressively shorter; sequences typically ended after 2–5 calls. Occasionally, the first call was split into a short call followed by a long one, after which the sequence proceeded normally. Under noise conditions, the overall sequence structure was preserved, although the first call was usually already significantly shorter than during baseline.

To compare individual calls across conditions, we selected pairs of noise and subsequent baseline sequences that shared the same structure—that is, a long first call followed by calls of decreasing duration. We then ranked the calls according to their position in the sequence and performed paired comparisons between calls of the same rank. Because baseline sequences generally contained more calls, we restricted the analysis to the number of calls present in the shorter sequence. Similarly, when comparing variables like total call duration, we used only as many baseline-calls as were present in the preceding noise sequence.

Change: We added now this description in the manuscript method section.

11) 294: Wilcoxon;

Comment: Because of the messed up text in the question, we are unable to understand the question accurately.

Change: None.

12) 583: Identification of responsive cells. Could the author elaborate on the advantage of using an iterative threshold value compared to a fixed one?

Comment: The iterative approach was designed to apply a conservative criterion for identifying non-responsive cells. Our goal was to ensure that, by the end of the procedure, the non-responsive group contained no neurons with responses exceeding 1.96 standard deviations from that group's mean.

We began with a single initial set (B_0) that included all neurons, treating them as non-responsive. After the first iteration, the classification produced two sets:

A_1 : neurons classified as responsive

B_1 : the updated non-responsive set, consisting of the previous set minus those now labeled responsive

However, because the standard deviation depends on which neurons remain in the pool, the updated non-responsive set B_1 might still contain neurons whose responses exceed the new 1.96-SD threshold. To address this, we recalculated the standard deviation of B_1 and repeated the classification.

By iterating this process until it no longer changed the composition of the non-responsive pool, we ensured that the final set contained only neurons meeting the criterion. At convergence, the final z-score threshold (1.96 standard deviations of the last non-responsive pool) can be converted back into the original response units and treated as a fixed detection threshold.

Change: None

REFERENCES

Rimehaug, A. E., Stasik, A. J., Hagen, E., Billeh, Y. N., Siegle, J. H., Dai, K., ... & Arkhipov, A. (2023). Uncovering circuit mechanisms of current sinks and sources with biophysical simulations of primary visual cortex. *elife*, 12, e87169.

Auksztulewicz, R., Rajendran, V. G., Peng, F., Schnupp, J. W. H., & Harper, N. S. (2023). Omission responses in local field potentials in rat auditory cortex. *BMC biology*, 21(1), 130.

Szymanski, F. D., Garcia-Lazaro, J. A., & Schnupp, J. W. (2009). Current source density profiles of stimulus-specific adaptation in rat auditory cortex. *Journal of neurophysiology*, 102(3), 1483-1490.